# Effects of Biophysical Factors on Light Use Efficiency at Multiple Time Scales in a Chinese Cork Oak Plantation Ecosystem

Xiang Gao [1,2,3], Jinsong Zhang [1,2,3,*], Jinfeng Cai [3], Ping Meng [1,2,3], Hui Huang [1,2,3] and Shoujia Sun [1,2,3]

[1] Key Laboratory of Tree Breeding and Cultivation, National Forestry and Grassland Administration, Research Institute of Forestry, Chinese Academy of Forestry, Beijing 100091, China; gaoxiang@caf.ac.cn (X.G.); mengping@caf.ac.cn (P.M.); huanghui@caf.ac.cn (H.H.); sunshj@caf.ac.cn (S.S.)

[2] Henan Xiaolangdi Forest Ecosystem National Observation and Research Station, Jiyuan 454650, China

[3] Collaborative Innovation Center of Sustainable Forestry in Southern China, Nanjing Forestry University, Nanjing 210037, China; caijinfeng1984@njfu.edu.cn

\* Correspondence: zhangjs@caf.ac.cn

**Abstract:** Light use efficiency (LUE) characterizes the efficiency of vegetation in converting photosynthetically active radiation (PAR) into biomass energy through photosynthesis and is a critical parameter for gross primary productivity (GPP) in terrestrial ecosystems. Based on the eddy covariance measurements of a Chinese cork oak plantation ecosystem in northern China, the temporal variations in LUE were investigated, and biophysical factors were examined at time scales ranging from hours to years. Our results show that diurnal LUE first increased sharply before 8:30 and then decreased gradually until 12:00, thereafter increasing gradually and reaching the maximum value at sunset during the growing season. The daily and monthly LUE first increased and then decreased within a year and showed a substantial drop around June. The annual LUE ranged from 0.09 to 0.17 g C mol photon$^{-1}$, and the multiyear mean maximal LUE was 0.30 g C mol photon$^{-1}$ during 2006–2019. Only GPP (positive) and clearness index (CI) (negative) had consistent effects on LUE at different time scales, and the effects of the remaining biophysical factors on LUE were different as the time scale changed. The effects of air temperature, vapor pressure deficit, precipitation, evaporative fraction, and normalized difference vegetation index on LUE were mainly indirect (via PAR and/or GPP). When CI decreased, an increased ratio of diffuse PAR to PAR produced a more uniform irradiance in the canopy, which ultimately resulted in a higher LUE. Due to climate change in our study area, the annual LUE may decrease in the future but improving management practices may slow or even reverse this trend in the annual LUE in the studied Chinese cork oak plantation.

**Keywords:** light use efficiency; temporal variation; biophysical control; Chinese cork oak plantation

## 1. Introduction

Light use efficiency (LUE) reflects the capacity of vegetation using photosynthetically active radiation (PAR) to absorb $CO_2$ in the atmosphere; it characterizes the efficiency of vegetation in converting PAR (solar energy) into organic matter (biomass energy) through photosynthesis and is a critical index of ecosystem productivity [1,2]. LUE directly connects PAR with ecosystem productivity, which provides a simple and feasible approach to estimate ecosystem productivity using PAR data [3,4]. Many LUE-based carbon exchange models can precisely estimate gross primary production (GPP) and net primary productivity (NPP) at different spatial and temporal scales, provided that a priori estimates of LUE are accurate [5,6]. Therefore, it is necessary to ascertain the spatiotemporal variations in LUE and the relationship between LUE and abiotic and biotic factors to improve our knowledge of terrestrial vegetation characteristics and the global carbon cycle.

Conceptually, two methods are used to calculate LUE. One can calculate LUE by dividing the GPP by the absorbed PAR. LUE calculated using this method is known as physiological LUE and is widely used in MODIS-based LUE models [7,8], in studies

ranging from leaf to canopy levels [5], and for determining physiological and biochemical parameters associated with photosynthesis [9]. The other method estimates LUE as the ratio of GPP to incoming PAR. LUE determined using this method is defined as ecological LUE and is applied in studies at the ecosystem level [2,10] and for eddy covariance (EC)-based LUE models to scale GPP measured using EC techniques up to the regional level [9,10]. In this study, we focus on LUE calculated using the second method (i.e., ecological LUE, hereafter simply referred to as LUE), which reflects the ecophysiological characteristics of vegetation and takes into account several ecosystem-level properties, such as the leaf area index, plant density, and aboveground biomass [11].

Over the past two decades, EC techniques have become the standard tool for measuring fluxes of energy, water vapor, $CO_2$, and $CH_4$ at a high temporal resolution between the atmosphere and terrestrial ecosystems [12]. Evapotranspiration and carbon fluxes estimated from satellite data need to be verified by the values measured from EC techniques [13,14], which are also used to evaluate the performance of MODIS-based LUE models [7,8]. Ecosystem GPP at scales ranging from hours to years can be accurately estimated from the data measured by the EC system [15]; hence, EC-based LUE is widely used in carbon exchange studies and has received widespread attention from ecologists [2,9,10]. Biophysical factors are often measured in parallel with EC-based LUE at well-instrumented sites [2,5,6], offering an excellent opportunity to assess their effects on LUE at different time scales.

Many studies have shown that vegetation properties (vegetation type [9], species composition [16], stand age [17], and canopy structure [18]), climate (solar radiation [19], air temperature ($T_a$) [20], precipitation (P) [3], and $CO_2$ concentration [21]), and management practices (pruning, thinning [22], fertilization [23], and irrigation [24]) determine the LUE variations in terrestrial ecosystems. In forest ecosystems, the LUE is generally greater in evergreen broadleaf forest than in other forests [25], and the spatial heterogeneity of LUE is mainly determined by the annual P and annual mean $T_a$ [3,20,26]. As a single species is usually used for plantations, the LUE is often lower in plantations than that in natural forests with a complex canopy [18], but management practices can compensate for this to a degree in the same region [24]. The effects of biophysical factors on LUE in plantations are similar to natural forests, that is, plant growth dominates the general trend in LUE and physical factors control the day-to-day variations in LUE. However, the effects of physical factors on temporal variations in LUE vary for different plantations [27]. For example, $T_a$ was the dominant influencing factor of the LUE in a coniferous plantation in Northeast China [3], and the LUE was insensitive to $T_a$ in an evergreen broadleaf plantations in southern China [28]. Tong et al. (2017) reported that LUE had a negative correlation with vapor pressure deficit (VPD) in a warm temperate mixed plantation in north China, but the LUE increased with increasing VPD in a rubber plantation in South China [27]. Knowledge of the temporal variation in LUE and its controlling factors is necessary for simulating and predicting energy fixation and the carbon cycle in the context of global climate change [11,29], but this knowledge is still scarce in many plantation ecosystems.

Plantation area ($8.00 \times 10^8$ ha) accounts for 36% of the total forest area in China and plays a key role in maintaining ecological security and promoting economic development [30]. As the LUE can be used to estimate the GPP, studying the LUE in plantations may help predict carbon sequestration and wood production [31]. Chinese cork oak (*Quercus variabilis* Bl.) is widely used in the reforested areas of lithoid hills because of its resilience to drought and barren soil, and it is used for more than 80% of soil and water conservation engineering projects in China [32]. In addition, Chinese cork oak is a species with high-quality biomass energy and is used to establish energy forests in northern China [33]. Chinese cork oak plantations have revealed the remarkable role of vegetation in afforested areas. In this study, we report a dataset of eddy covariance measurements for a Chinese cork oak plantation created during 2006–2019 in northern China. The specific objectives of our study were to (1) examine the temporal variations in LUE at scales ranging from hours to years and (2) investigate the effect of biophysical factors on LUE at different time scales.

As LUE was the ratio of GPP to PAR in this study, it was hypothesized that the effects of other biophysical factors on LUE were mainly via the GPP and PAR at various time scales.

## 2. Materials and Methods

### 2.1. Study Area

This study was conducted at the Henan Xiaolangdi Forest Ecosystem National Observation and Research Station (35°01′ N, 112°28′ E, a.s.l. 410 m), located at a lithoid hilly area adjacent to (south of) the Taihang Mountains in northern China (Figure 1). This study station experiences a warm temperate continental monsoon climate with an annual mean temperature of 13.4 °C. The annual precipitation is 642 mm, 68.3% of which occurs from June to September. The soil is mainly composed of brown loam, with a thin layer approximately 40 cm deep and poor nutrient contents [34].

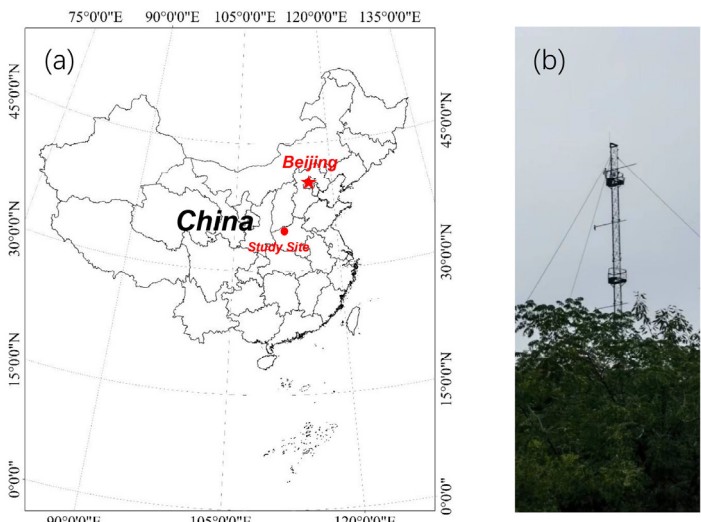

**Figure 1.** Location of the study site (**a**) and the observation tower (**b**).

The Chinese cork oak plantation under study (7210 ha) is 47 years old with a density of 1905 stems ha$^{-1}$, an average diameter at breast height of 11.3 cm, and a canopy height of approximately 10.2 m. Cork oak accounts for more than 80% of the plantation area, and the remaining includes arborvitae (*Platycladus orientalis*) and black locust (*Robinia pseudoacacia*). The understory is mainly composed of sour jujube (*Ziziphus jujuba* Mill. var. *inermis* (Bunge) Rehd.) and bunge hackberry (*Celtis bungeana* Bl.) [34]. The Chinese cork oak plantation growing season is from April to October, and the canopy closure of this plantation occurs from May to September.

### 2.2. Measurement

The latent heat flux (LE), sensible heat flux (H), and net ecosystem exchange (NEE) were measured using an EC system comprising a three-dimensional sonic anemometer (CSAT3, Campbell Scientific Inc., Logan, UT, USA) and an open-path infrared $CO_2/H_2O$ analyzer (LI-7500, Li-COR Inc., Lincoln, NE, USA). The EC system was installed at a height of 30 m on a tower situated at the center of the plantation. Raw data were collected at 10 Hz and stored in a data logger (CR5000, Campbell Scientific Inc.). The distance from the tower to the nearest boundary of the plantation is approximately 6 km, ensuring that the measured signal originated from the plantation.

Air temperature ($T_a$) and relative humidity ($H_a$) were measured using seven $T_a$–$H_a$ sensors (HMP45C, Vaisala Co., Ltd., Helsinki, Finland) at heights of 5, 8, 11, 14, 18, 26, and 32 m. A net radiometer (CNR1, Kipp and Zonen B.V., Delft, The Netherlands) was used to collect solar radiation ($S_r$) and net radiation at a height of 17 m. The soil heat flux at a 5 cm depth was measured using three soil heat flux plates (HFP01SC, Hukseflux

B.V., Delft, The Netherlands) around the tower. The instruments were connected to a data logger (CR1000, Campbell Scientific Inc.), and the mean data were stored at 30 min intervals. Photosynthetically active radiation (PAR) was collected at 30 min intervals using a PAR sensor (LI190SB, Li-COR Inc.) connected to a separate data logger (CR1000, Campbell Scientific Inc.) located 750 m from the tower at a standard meteorological station. The annual precipitation (P), annual $S_r$, and annual mean $T_a$ during 1960–2019 were collected from the Jiyuan National Meteorological Station located 15 km from the tower. The trends in the annual $S_r$, annual mean $T_a$, and annual P during 1960–2019 are shown in Figure 2. The monthly mean normalized difference vegetation index (NDVI), an indicator of vegetation characteristics, was downloaded from the World Meteorological Organization (http://climexp.knmi.nl/start.cgi, (accessed on 15 May 2021)).

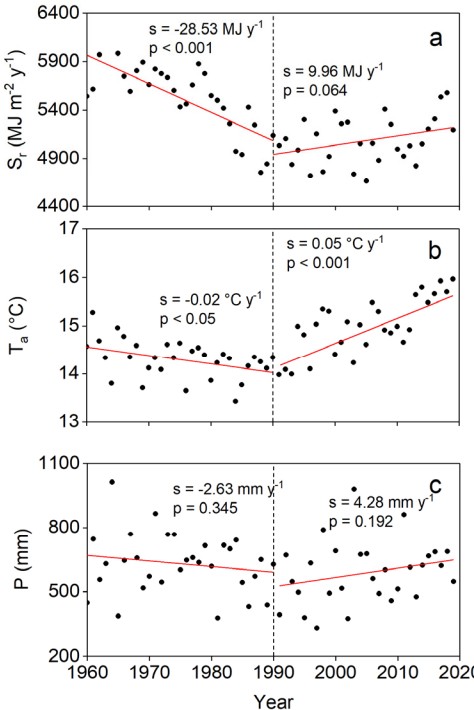

**Figure 2.** General trends in (**a**) annual solar radiation ($S_r$), (**b**) annual mean air temperature ($T_a$), and (**c**) annual precipitation (P) in our study area during 1960–2019. The *p* and s values are the significance levels and slopes of the red lines.

*2.3. Data Proceeding*

2.3.1. Flux Data

Eddypro (7.0.7, Li-COR Inc.) software was used for calibration and quality control of the 10 Hz raw data and generated 30 min data. The correction of the turbulent flux data included planar fit correction, sonic temperature correction, density fluctuation correction, and spectral correction. The calculated fluxes data were quality controlled by deleting those data with a quality flag of "2". The friction velocity ($u^*$) thresholds for this site across the years, ranging from 0.12 m s$^{-1}$ to 0.26 m s$^{-1}$, were calculated using the REddyProc R package based on the moving point test method (https://cran.rstudio.com/web/packages/REddyProc/index.html, (accessed on 9 March 2022)). Furthermore, 46% of the NEE, 42% of the LE, and 32% of the H values were rejected because of quality control, $u^*$ filtering, and equipment failure. The energy budget closure of the post-processed 30 min data was 80%. Gap filling was conducted using the marginal distribution sampling method, and the gross primary production (GPP) was calculated using NEE partitioning based on the night-time flux partitioning method in the REddyProc R package. The valid original data were used for statistical analyses at the 30 min time scale, and the filled data were used at other time scales.

### 2.3.2. Light Use Efficiency

In this study, the light use efficiency (LUE) was calculated as follows:

$$LUE = \frac{GPP}{PAR} \tag{1}$$

where *GPP* is the gross primary production (µmol m$^{-2}$ s$^{-1}$), and *PAR* is the photosynthetically active radiation (µmol m$^{-2}$ s$^{-1}$).

### 2.3.3. Evaporative Fraction

The evaporative fraction (EF) is an indicator of vegetation growth conditions and can be used to indicate the vegetation moisture status during the canopy closure period. The *EF* was calculated as follows [31]:

$$EF = \frac{LE}{LE + H} \tag{2}$$

where *LE* is the latent heat flux (W m$^{-2}$) and *H* is the sensible heat flux (W m$^{-2}$).

### 2.3.4. Clearness Index

The clearness index (CI) is an indicator of clouds and/or aerosols in the sky: when it is closer to 0, more clouds and/or aerosols are in the sky, and when it is closer to 1, fewer clouds and/or aerosols are in the sky. The *CI* was calculated as follows [35]:

$$CI = \frac{S_r}{S_e} \tag{3}$$

where $S_r$ is the solar radiation (W m$^{-2}$) and $S_e$ is the extraterrestrial irradiance at a plane parallel to the Earth's surface (W m$^{-2}$), calculated as follows:

$$S_e = S_{sc}[1 + 0.033\cos{(360t_d/365)}]\sin{\beta} \tag{4}$$

$$\sin{\beta} = \sin{\varphi}\sin{\delta} + \cos{\varphi}\cos{\delta}\cos{\omega} \tag{5}$$

where $S_{sc}$ is the solar constant (1370 W m$^{-2}$), $t_d$ is the day of the year, $\beta$ is the solar elevation angle, $\varphi$ is the local latitude, $\delta$ is the declination of the sun, and $\omega$ is the hour angle.

### 2.3.5. PAR Partitioning

The diffuse PAR ($PAR_f$) and direct PAR ($PAR_r$) were calculated using these equations [36]:

$$TD_f = \frac{[1 + 0.3(1 - q^2)]q}{1 + (1 - q^2)\cos^2(\pi/2 - \beta)\cos^3\beta} \tag{6}$$

$$q = (S_f/S_e)/CI \tag{7}$$

$$PAR_f = PAR \cdot TD_f \tag{8}$$

$$PAR_r = PAR - PAR_f \tag{9}$$

where $TD_f$ is the ratio of $PAR_f$ to *PAR* and $S_f$ is the total diffuse radiation (visible plus near-infrared) received by a horizontal plane on the Earth's surface. The detailed calculation method for the $S_f/S_e$ can be found in Appendix A.

### *2.4. Statistical Analyses*

### 2.4.1. Path Analyses

Path analyses can decompose the correlation coefficient into direct and indirect coefficients on the basis of correlation and regression analyses to reveal the relative importance of various independent variables to a dependent variable. The calculated process supposes

there are $n$ independent variables, $x_1, x_2, \ldots, x_n$, and one dependent variable, $y$. We define the correlation coefficient between each independent variable as $r_{ij}$ ($i = 1, 2, \ldots, n; j = 1, 2, \ldots, n$), and then a standardized normal equation for solving the path coefficients is constructed using a simple correlation between $r_{ij}$ and $y$.

$$\begin{cases} r_{11}{\cdot}\rho_1 + r_{12}{\cdot}\rho_2 + \cdots + r_{1n}{\cdot}\rho_n = r_{1y} \\ r_{21}{\cdot}\rho_1 + r_{22}{\cdot}\rho_2 + \cdots + r_{2n}{\cdot}\rho_n = r_{2y} \\ \qquad\qquad\vdots \\ r_{n1}{\cdot}\rho_1 + r_{n2}{\cdot}\rho_2 + \cdots + r_{nn}{\cdot}\rho_n = r_{ny} \end{cases} \tag{10}$$

where $\rho_1, \rho_2, \ldots, \rho_n$ are the direct path coefficients from $x_i$ to $y$. Define the matrix Equation (10) as **r**, and then $\rho_i$ ($i = 1, 2, \ldots, n$) can be derived by calculating the inverse matrix (**c**) of **r**. It can be expressed as follows:

$$\begin{bmatrix} \rho_1 \\ \rho_2 \\ \vdots \\ \rho_n \end{bmatrix} = \begin{bmatrix} c_{11}c_{12}c_{13} \cdots c_{1n} \\ c_{21}c_{22}c_{23} \cdots c_{2n} \\ \vdots\vdots\vdots \\ c_{n1}c_{n2}c_{n3} \cdots c_{nn} \end{bmatrix} \begin{bmatrix} r_{1y} \\ r_{2y} \\ \vdots \\ r_{ny} \end{bmatrix} \tag{11}$$

The indirect path coefficient can be calculated as the product of the correlation coefficient and the direct path coefficient [37]. In this study, the influence of biophysical factors on the LUE was determined with path analyses using SAS software (Version 9, SAS Institute, Cary, NC, USA) at multiple time scales.

### 2.4.2. Other Statistical Analyses

Correlation analysis and stepwise regression among the biophysical factors and LUE were performed using SPSS software (Version 18, SPSS Inc., Chicago, IL, USA). This software was also used to calculate the variance inflation factor among the influencing factors.

## 3. Results

### 3.1. LUE at the 30 min Time Scale and Influencing Factors

Monthly mean patterns of diurnal PAR, GPP, and LUE during the growing seasons (April–October) of the Chinese cork oak plantation in 2018 and 2019 are presented in Figure 3. For each month, the monthly mean diurnal PAR exhibited a regular "∧" pattern during the daytime, peaking at around 12:00 (Figure 3a). Since the diurnal course of the GPP was mainly determined by that of the PAR, the monthly mean diurnal variation in the GPP was similar to that of the PAR, and the monthly mean diurnal GPP was close to its maximum value from 9:00 to 14:00 in all months (Figure 3b). The monthly mean diurnal LUE first increased sharply and reached a peak value at around 8:30, and gradually decreased until 12:00. It increased gradually in the afternoon and reached its maximum value at sunset (Figure 3c). Compared to other months, the monthly mean diurnal LUE values were lowest at around 12:00 in June (Figure 3c).

Because of the dramatic changes in LUE at sunrise/sunset (Figure 3c) and the effect of vegetation growth on the LUE, the measured 30 min data from 10:00 to 14:00 during May–September (canopy closure period) in 2018 and 2019 were used to assess the effects of biophysical factors on the LUE at the 30 min time scale (Figure 4). As shown in Figure 4a, all the biophysical factors influenced the LUE at a significance level of $p < 0.001$. The LUE was positively correlated with the EF and GPP, with correlation coefficient ($r$) values of 0.48 and 0.68, respectively, and the remaining factors (PAR, CI, $T_a$, and VPD) were negatively correlated with the LUE, with $r$ values ranging from $-0.39$ to $-0.67$. However, a path analysis between the LUE and the biophysical factors showed that the effects of PAR and GPP on the LUE were mainly direct; the effects of the remaining factors (EF, CI, $T_a$, and VPD) on the LUE were mainly indirect (via PAR and/or GPP); the direct effect of the CI on the LUE was $-0.19$ (Figure 4b).

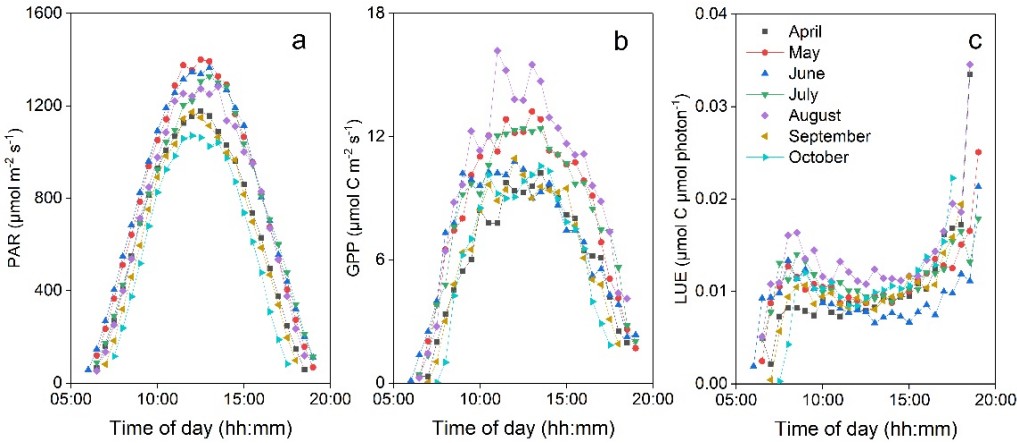

**Figure 3.** Monthly mean diurnal course (PAR > 50 μmol m$^{-2}$ s$^{-1}$) of (**a**) photosynthetically active radiation (PAR), (**b**) gross primary productivity (GPP), and (**c**) light use efficiency (LUE) in the Chinese cork oak plantation during the growing seasons of 2018 and 2019.

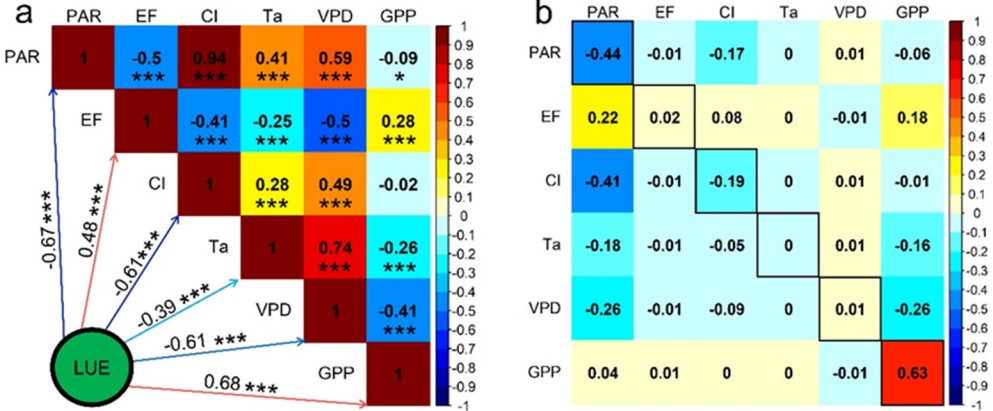

**Figure 4.** Correlation analysis (**a**) and path analysis (**b**) among 30 min biophysical factors and LUE around noon (10:00–14:00) during May–September in 2018 and 2019 in the Chinese cork oak plantation. * Significant at $p < 0.05$; *** significant at $p < 0.001$.

### 3.2. LUE at the Daily Time Scale and Influencing Factors

Seasonal variations in the daily PAR, GPP, and LUE in 2018 and 2019 are shown in Figure 5. The daily PAR displayed a parabolic trend, peaked in late June, and sharply decreased on rainy/cloudy days (Figure 5a). Two peaks in the general trend in the daily GPP appeared in early May and mid-August. The daily GPP dramatically decreased on rainy/cloudy days and subsequently increased sharply on sunny days (Figure 5b). Similar to the daily GPP, the daily LUE generally first increased and then decreased within a year (Figure 5c).

We selected the measured daily data from May to September in 2018 and 2019 to assess the effects of biophysical factors on the LUE at a daily time scale (Figure 6). The results of the correlation analysis between the biophysical factors and the LUE are shown in Figure 6a. All biophysical factors influenced the LUE at a significance level of $p < 0.001$; the *r* values between the EF and GPP and the LUE were 0.53 and 0.56, respectively, and the *r* values between the LUE and the PAR, CI, Ta, and VPD were −0.66, −0.65, −0.37, and −0.36, respectively. The effects of the PAR and GPP on the LUE were mainly direct; those of the EF, CI, Ta, and VPD on the LUE were mainly indirect, via PAR and/or GPP; the direct effect of the CI on LUE was −0.17 (Figure 6b).

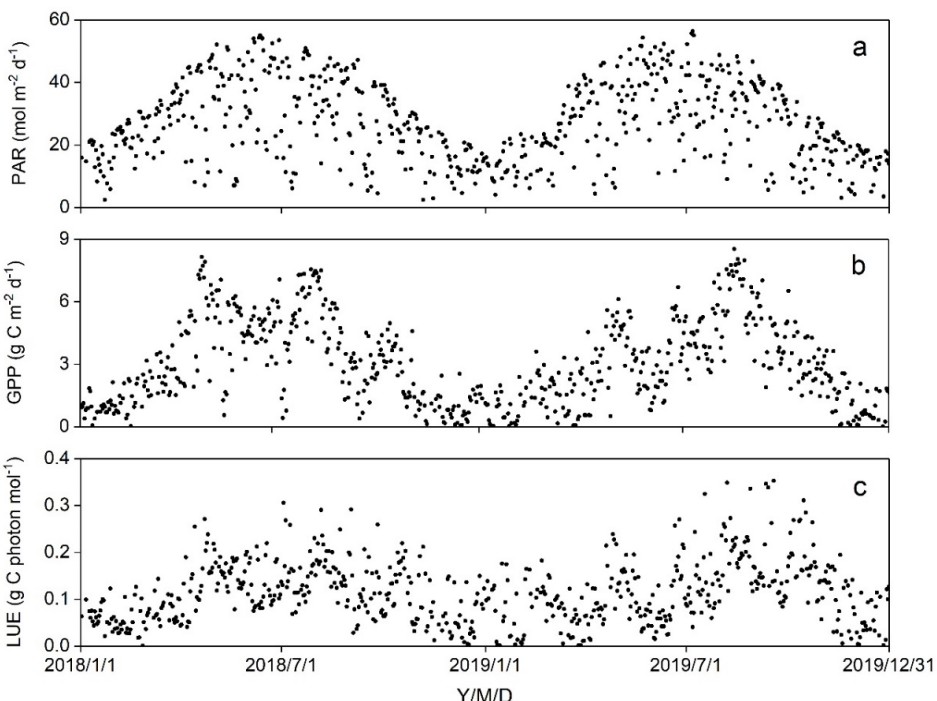

**Figure 5.** Seasonal variations in the daily PAR (**a**), GPP (**b**), and LUE (**c**) in the Chinese cork oak plantation in 2018 and 2019.

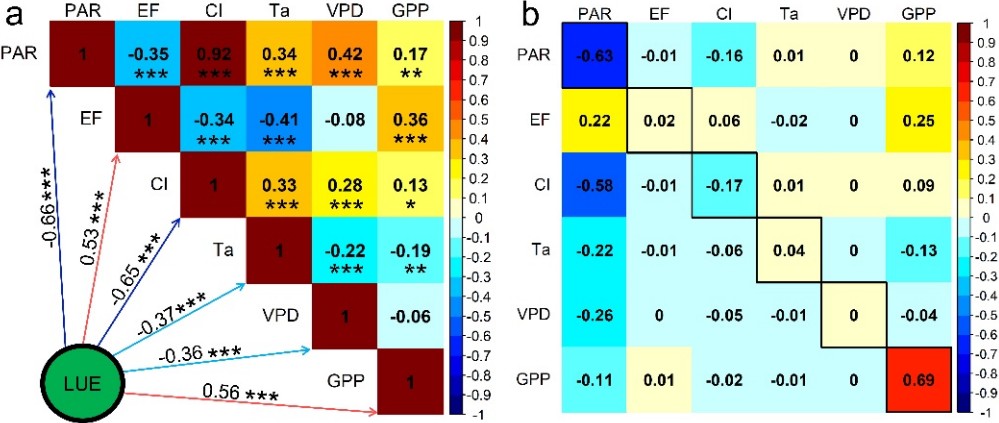

**Figure 6.** Correlation analysis (**a**) and path analysis (**b**) among daily biophysical factors and LUE during May–September in 2018 and 2019 in the Chinese cork oak plantation. * Significant at $p < 0.05$; ** significant at $p < 0.01$; *** significant at $p < 0.001$.

The mechanisms by which the CI influences LUE are shown in Figure 7. In this study, the GPP rapidly increased when CI ≤ 0.23, slightly increased when $0.23 < CI < 0.57$, and rapidly decreased when CI ≥ 0.57 (Figure 7a). The $TD_f$ was relatively unresponsive to the CI when CI ≤ 0.23, and rapidly decreased when CI > 0.23 (Figure 7b). The $PAR_r$ was relatively unresponsive to the CI when CI ≤ 0.23, and rapidly increased when CI > 0.23 (Figure 7c). The relationship between the $PAR_f$ and CI was a downward parabola, peaking at CI = 0.42 (Figure 7d). The GPP was insensitive to the $PAR_r$ (Figure 7e) but increased with increasing $PAR_f$ (Figure 7f), indicating that the $PAR_f$ affected the GPP more than the $PAR_r$ in this study. The relationships between the $PAR_f$ and GPP and between the CI and $PAR_f$ resulted in the relationship between the GPP and CI, and the negative relationship between the CI and LUE resulted from the different effects of the CI on the GPP and PAR.

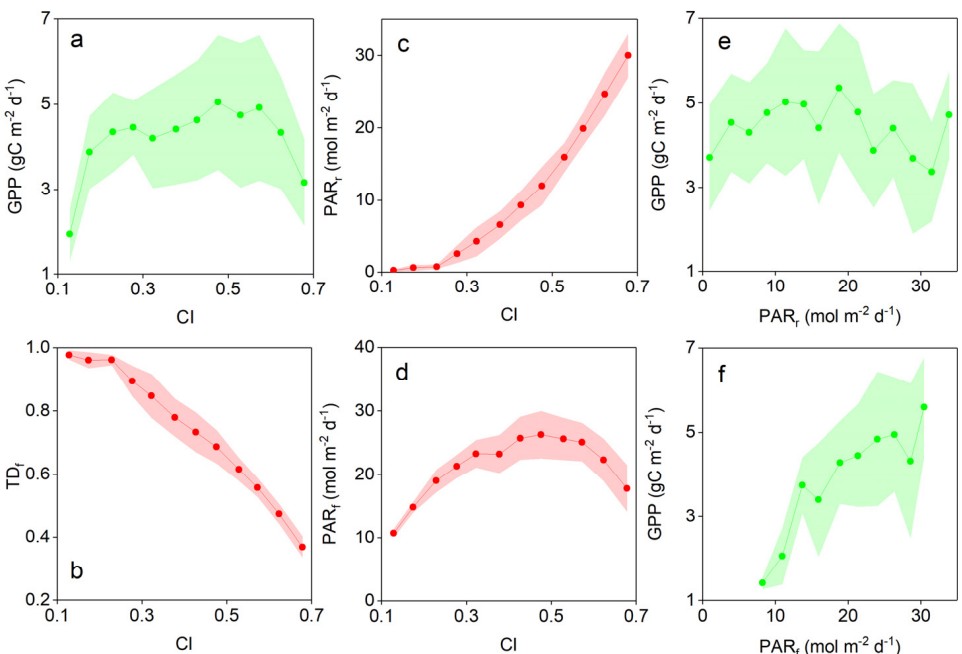

**Figure 7.** Relationship between the CI and (**a**) GPP; (**b**) the ratio of diffuse PAR (PAR$_f$) to PAR (TD$_f$), (**c**) direct PAR (PAR$_r$), and PAR$_f$ (**d**); and the effects of PAR$_r$ (**e**) and PAR$_f$ (**f**) on GPP during May–September in 2018 and 2019 in the Chinese cork oak plantation. Daily GPP (**a**), TD$_f$ (**b**), PAR$_r$ (**c**), and PAR$_f$ (**d**) were bin-averaged into 0.05 CI increments. Daily GPP was bin-averaged into 2.50 PAR$_r$ (**e**) and 2.50 PAR$_f$ (**f**).

### 3.3. LUE at the Monthly Time Scale and Influencing Factors

Seasonal variations in the monthly biophysical factors and LUE from 2006 to 2019 are shown in Figure 8. The monthly PAR exhibited a general trend similar to the monthly Ta; the former peaked in May, and the latter peaked in July (Figure 8a,b). The monthly VPD generally increased during January–June and sharply decreased in July, indicating the beginning of summer monsoon influence (Figure 8c). The monthly P was relatively greater in June–September because of the summer monsoon (Figure 8d). The monthly EF was lower in the non-growing season, generally increased during April–August, and then declined (Figure 8e). The monthly CI had no major seasonal variations and was relatively lower during June–September (Figure 8f). The monthly NDVI sharply increased in April, remained relatively stable from May to September, and decreased from October (Figure 8g). The monthly GPP exhibited general trends similar to the monthly NDVI, but it increased from March and decreased from September (Figure 8h). The monthly LUE generally first increased and then decreased in the growing season, and the mean monthly LUE was lower than 0.1 g C mol photon$^{-1}$ in the non-growing season (Figure 8i).

The results of the correlation analysis among the monthly biophysical factors and LUE is shown in Figure 9, where only the CI had a negative effect on LUE, with an *r* value of −0.52 at a significance level of $p < 0.001$. The PAR ($r = 0.28$) and VPD ($r = 0.18$) influenced the LUE at significance levels of $p < 0.01$ and $p < 0.05$, respectively. The remaining factors (T$_a$, P, EF, NDVI, and GPP) were positively correlated with LUE, with *r* values ranging from 0.53 to 0.83 at a significance level of $p < 0.001$.

As shown in Figure 10a, the values of the variance inflation factors for PAR, T$_a$, EF, and NDVI were greater than five, indicating that multicollinearity existed in the monthly biophysical factors, and the path analysis was unsuitable for assessing the independent effect of each biophysical factor on the LUE. A partial correlation analysis between the biophysical factors and the LUE showed that the monthly PAR, CI, and GPP influenced the LUE at a significance level of $p < 0.001$, with partial correlation values of −0.61, −0.33, and 0.90, respectively. The partial correlation coefficient values between the monthly T$_a$, VPD,

P, EF, and NDVI and the LUE were approximately zero at a significance level of $p > 0.05$ (Figure 10b). Figure 10c shows the performance of the stepwise regression model, where the monthly estimated LUE ($LUE_e$) was close to the measured LUE ($LUE_m$), with an $R^2$ value of 0.93 at a significance level of $p < 0.001$.

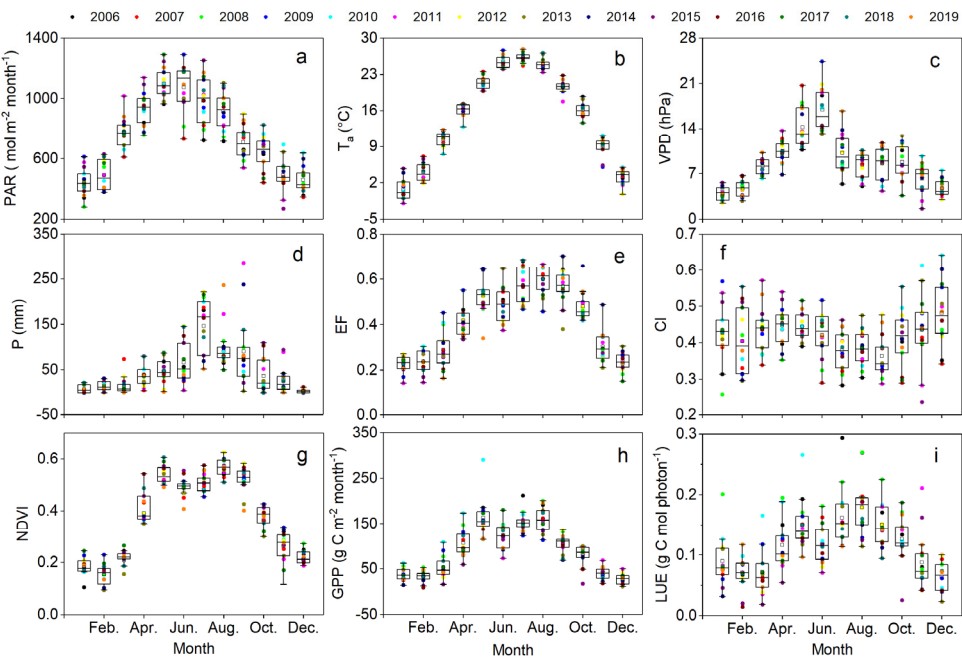

**Figure 8.** Seasonal variations in monthly PAR (**a**), $T_a$ (**b**), (**c**) vapor pressure deficit (VPD), P (**d**), (**e**) evaporative fraction (EF), CI (**f**), NDVI (**g**), GPP (**h**), and LUE (**i**) in 2006–2019 in the Chinese cork oak plantation.

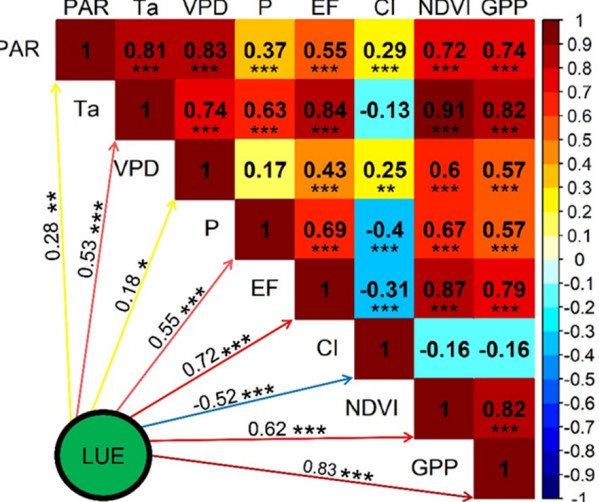

**Figure 9.** Correlation analysis among monthly biophysical factors and LUE in 2006–2019 in the Chinese cork oak plantation. * Significant at $p < 0.05$; ** significant at $p < 0.01$; *** significant at $p < 0.001$.

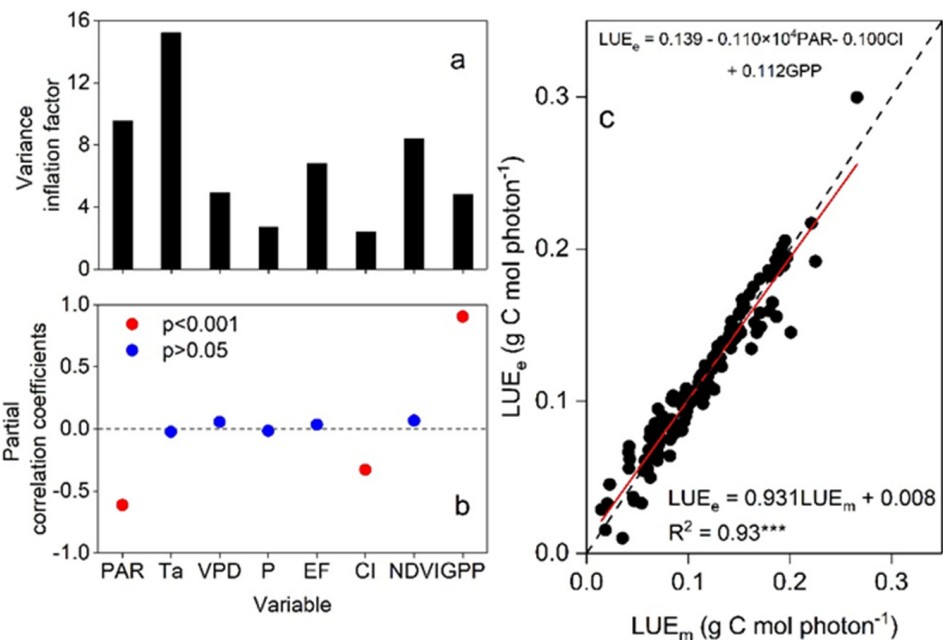

**Figure 10.** Variance inflation factors among monthly influencing factors (**a**), partial correlation coefficients between monthly biophysical factors and LUE (**b**), stepwise regression among monthly biophysical factors and LUE (**c**) in 2006–2019 in the Chinese cork oak plantation. *** significant at $p < 0.001$.

### 3.4. LUE at the Annual Time Scale and Influencing Factors

The annual variations in the biophysical factors, LUE, and $LUE_{max}$ during 2006–2019 are shown in Table 1. The annual PAR, $T_a$, VPD, and P ranged from 7670.72 to 9807.15 mol m$^{-2}$, from 13.81 to 15.51 °C, from 7.98 to 10.04 hPa, and from 456.40 to 860.00 mm, respectively. The annual EF, CI, NDVI, and GPP ranged from 0.38 to 0.45, from 0.36 to 0.45, from 0.34 to 0.40, and from 842.37 to 1426.12 g C m$^{-2}$, respectively. The annual LUE ranged from 0.09 to 0.17 g C mol photon$^{-1}$, with a mean value of 0.12 g C mol photon$^{-1}$. The 8-day mean maximal LUE was regarded as the $LUE_{max}$ to reduce the uncertainty caused by the noise data in each year. The annual $LUE_{max}$ ranged from 0.21 to 0.41 g C mol photon$^{-1}$, with a mean value of 0.30 g C mol photon$^{-1}$.

The results of the correlation analysis among the annual biophysical factors and LUE are shown in Figure 11a, where the annual GPP ($r = 0.90$), PAR ($r = -0.70$), and CI ($r = -0.60$) influenced the LUE at significance levels of $p < 0.001$, $p < 0.01$, and $p < 0.05$, respectively. The remaining factors ($T_a$, VPD, P, EF, and NDVI) did not influence the LUE because the significance level was $p > 0.05$. The effects of the GPP on the LUE were mainly direct; those of the PAR and CI on the LUE were mainly indirect, via the other two biophysical factors; the direct effect of the CI on was $-0.24$ (Figure 11b). As shown in Figure 1, the annual Sr increased by 9.96 MJ y$^{-1}$ ($p = 0.064$), the annual mean $T_a$ increased by 0.05 °C y$^{-1}$ ($p < 0.001$), and the annual P increased by 4.28 mm y$^{-1}$ ($p = 0.192$) after 1990 in the study area. Under these trends of climate change, the annual LUE of the Chinese cork oak plantation might decrease in the future, based on the negative effect of the annual PAR, $T_a$, and P on the annual LUE (Figure 11a).

**Table 1.** Annual photosynthetically active radiation (PAR), mean air temperature ($T_a$), mean vapor pressure deficit (VPD), precipitation (P), evaporative fraction (EF), clearness index (CI), normalized difference vegetation index (NDVI), gross primary productivity (GPP), light use efficiency (LUE), and maximal LUE ($LUE_{max}$) in the Chinese cork oak plantation during 2006–2019.

| Year | PAR (mol m$^{-2}$) | $T_a$ (°C) | VPD (hPa) | P (mm) | EF | CI | NDVI | GPP (g C m$^{-2}$) | LUE (g C mol photon$^{-1}$) | $LUE_{max}$ (g C mol photon$^{-1}$) |
|---|---|---|---|---|---|---|---|---|---|---|
| 2006 | 8048.67 | 15.03 | 8.49 | 559.70 | 0.45 | 0.37 | 0.37 | 1217.79 | 0.15 | 0.36 |
| 2007 | 7755.90 | 15.11 | 9.17 | 490.90 | 0.43 | 0.36 | 0.37 | 1036.66 | 0.13 | 0.31 |
| 2008 | 7670.72 | 14.64 | 9.19 | 600.10 | 0.42 | 0.36 | 0.37 | 1335.05 | 0.17 | 0.41 |
| 2009 | 9451.69 | 14.52 | 9.01 | 456.40 | 0.43 | 0.45 | 0.38 | 1090.35 | 0.12 | 0.36 |
| 2010 | 9212.84 | 14.26 | 8.13 | 512.10 | 0.43 | 0.44 | 0.37 | 1426.12 | 0.15 | 0.33 |
| 2011 | 9200.02 | 13.81 | 8.39 | 860.00 | 0.39 | 0.43 | 0.37 | 1059.85 | 0.12 | 0.27 |
| 2012 | 9792.10 | 14.38 | 9.02 | 613.30 | 0.41 | 0.45 | 0.38 | 1073.21 | 0.11 | 0.28 |
| 2013 | 8902.24 | 15.04 | 8.79 | 474.60 | 0.40 | 0.42 | 0.34 | 981.28 | 0.11 | 0.25 |
| 2014 | 9005.00 | 15.51 | 8.78 | 621.10 | 0.45 | 0.42 | 0.38 | 869.41 | 0.10 | 0.23 |
| 2015 | 9807.15 | 15.02 | 8.35 | 664.80 | 0.38 | 0.43 | 0.39 | 842.37 | 0.09 | 0.21 |
| 2016 | 9102.98 | 15.04 | 7.98 | 684.70 | 0.40 | 0.42 | 0.40 | 950.38 | 0.10 | 0.33 |
| 2017 | 9596.15 | 15.32 | 8.87 | 620.10 | 0.38 | 0.45 | 0.39 | 1156.96 | 0.12 | 0.38 |
| 2018 | 9527.76 | 15.12 | 8.87 | 686.20 | 0.41 | 0.45 | 0.38 | 1090.89 | 0.11 | 0.23 |
| 2019 | 9249.70 | 15.45 | 10.04 | 546.60 | 0.38 | 0.44 | 0.36 | 1038.20 | 0.11 | 0.29 |
| Mean | 9023.07 | 14.88 | 8.79 | 599.33 | 0.41 | 0.42 | 0.38 | 1083.47 | 0.12 | 0.30 |

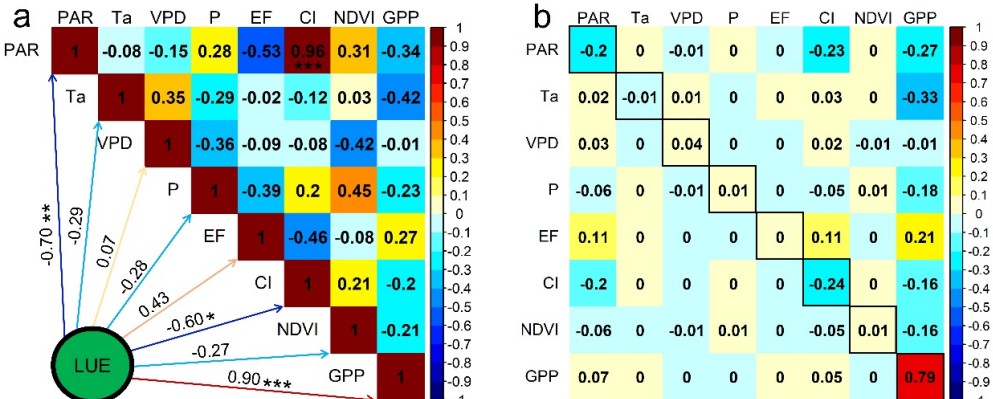

**Figure 11.** Correlation analysis (**a**) and path analysis (**b**) among annual biophysical factors and LUE in 2006–2019 in the Chinese cork oak plantation. * Significant at $p < 0.05$; ** significant at $p < 0.01$; *** significant at $p < 0.001$.

## 4. Discussion

### 4.1. Variations in LUE at Different Time Scales

The monthly mean hourly LUE increased sharply and reached a peak value in the morning at our study site (Figure 3c). We suspect that the following two factors are responsible for this phenomenon: (1) dark reaction lags behind the light reaction in the process of photosynthesis during the transition from dark to light, resulting in an increasing LUE after sunrise; (2) stomatal conductance increases sharply and reaches a peak value at around 8:30; thereafter, it decreases gradually in the morning [38,39], and the LUE increases with stomatal conductance [31]. Fei et al. (2018) found that the LUE decreased in a tropical rainforest ecosystem in the morning in southern China, and no major change in the LUE was observed in a rainfed spring maize field in the morning on the Loess Plateau, China [36]. The monthly mean diurnal course of LUE in the morning varies in different ecosystems, which could be due to different vegetation characteristics and environmental factors in different ecosystems. The monthly mean diurnal LUE increased gradually in the afternoon in this study (Figure 3c), which is consistent with the results in different terrestrial

ecosystems (e.g., meadow [13], forest [5], and cropland [36]). The main reasons for this phenomenon are (1) the vegetation photosynthesis consists of two relatively independent processes, i.e., light reaction and dark reaction; the former and latter absorb solar energy and $CO_2$, respectively; (2) the vegetation can use relatively more chemical energy accumulated by earlier light reactions to absorb $CO_2$ during the transition from light to dark, resulting in a slower decrease in the GPP than the PAR in the afternoon [36]. The monthly mean diurnal course of LUE in the afternoon is similar in different ecosystems, indicating a common trend in the LUE increasing in the afternoon among different terrestrial ecosystems.

The general trends in the daily and monthly LUE first increased and then decreased within a year in the Chinese cork oak plantation (Figures 5c and 8g), which agree with the results in temperate, subtropical, and tropical forests [5,28], because the vegetation dynamics and activity determined the general trends in the daily and monthly LUE [36]. Consistent with previous studies [5,36], day-to-day fluctuations were observed in the seasonal pattern of the daily LUE in this study; this is because the day-to-day fluctuations of the seasonal variations in the daily LUE were mainly controlled by meteorological factors [36]. Like the daily GPP, the daily LUE generally decreased between early May and mid-August in 2018 and 2019, and the mean monthly GPP and LUE were lower in June than in May and July, indicating that the decrease in the LUE in early summer is the norm in the Chinese cork oak plantation. The main reason for this phenomenon is that monthly VPD and EF in June were relatively higher and lower in the canopy closure period, respectively (Figure 8c,e), indicating that drought occurred in the same period, which inhibited vegetation activity. The start time for the phenomenon was determined by the water balance of the previous year and P during January–April of the current year, and the phenomenon ended with the beginning of the rainy season in the current year.

During 2006–2019, the annual LUE was highest in 2008 (0.17 g C mol photon$^{-1}$) and lowest in 2015 (0.09 g C mol photon$^{-1}$), indicating that significant interannual LUE variation occurred in the Chinese cork oak plantation; this finding is consistent with previous studies conducted in different ecosystems [5,12,27]. The mean annual LUE in the Chinese cork oak plantation was different from that in different ecosystems [10,20,26]; for example, higher than that in a grassland (0.02 g C mol photon$^{-1}$) and lower than that in an evergreen broadleaf forest (0.24 g C mol photon$^{-1}$) in Australia [10]. This indicates that the annual LUE varies in different ecosystems due to different vegetation characteristics and natural conditions. In addition, the mean annual LUE was 0.11 g C mol photon$^{-1}$ in a plantation oak woodland in south-eastern England [40], similar to our results. Similar vegetation types are likely responsible for this phenomenon.

Like the annual LUE, the annual LUE$_{max}$ showed great interannual variation (0.21–0.41 g C mol photon$^{-1}$) in the Chinese cork oak plantation, due to the different hydrothermal conditions in each year, and agreeing well with previous findings obtained in various ecosystems [2,5]. Therefore, the annual LUE$_{max}$ should be determined based on multi-year observation data when a carbon exchange model based on LUE is used to estimate the GPP at the regional scale. Zhang and Zhu (2018) [27], using measured net primary productivity, meteorological data, and remote sensing technology, estimated that the annual LUE$_{max}$ ranged from 0.06 to 0.55 g C mol photon$^{-1}$ in deciduous broad-leaved forests of China, and our results fall within this range.

### 4.2. Effect of Biophysical Factors on LUE at Different Time Scales

At the 30 min time scale, the measured 30 min data around noon in the canopy closure period of 2018 and 2019 were used to assess the effects of biophysical factors on the LUE in our study. The CI was calculated using Equation (3), and the ratio of PAR to $S_r$ was quite constant [41,42], resulting in a high *r* value between the PAR and CI (0.94, Figure 4a). As observed in previous studies [43,44], the CI was negatively correlated with the LUE in the Chinese cork oak plantation. The effects of the EF, $T_a$, and VPD on the LUE were mainly indirect, via PAR and GPP (Figure 4b); the $T_a$ and VPD had negative effects on the GPP, due to higher $T_a$ and VPD resulting in stomatal closure around noon (Figure 4a),

which agrees well with previous studies [42,44]. Many studies have shown that water stress can inhibit vegetation photosynthesis in various ecosystems [3,42]; hence, a positive relationship between the EF and LUE existed in our study (Figure 4a), as the EF is an indicator of vegetation moisture conditions in the canopy closure period. In addition, the effects of the PAR, CI, $T_a$, and VPD on the EF and GPP were negative (Figure 4a), illustrating that the light saturation phenomenon caused by water stress often occurred around noon during the canopy closure period in the Chinese cork oak plantation.

The effects of biophysical factors on the LUE at the daily time scale in our study were also found in rainforest and savanna ecosystems [3], because measurement data in the canopy closure period were used to eliminate the influence of vegetation growth on the LUE. The biophysical factors had similar effects on the LUE at the 30 min and daily time scales in the Chinese cork oak plantation (Figures 4a and 6a), but the effects of the PAR, CI, and VPD on the GPP were different at the two time scales, as the light saturation around noon could not determine the vegetation photosynthesis throughout the day. The difference in penetration of $PAR_r$ and $PAR_f$ in the canopy and the different effect of the $PAR_r$ and $PAR_f$ on the GPP were the root causes of the LUE increasing with CI decreasing, due to the following factors: the $PAR_r$ could only illuminate a fraction of leaves in the upper canopy, which were often light-saturated [35]; the $PAR_f$ could provide energy for most leaves shaded by upper leaves in the canopy, which were generally light-limited [20,44]. With increasing clouds and/or aerosols in the sky, the PAR received by the canopy decreased, but the $TD_f$ increased, producing a more uniform irradiance in the canopy. This led to a faster decrease in the PAR than the GPP, which resulted in an increase in the LUE [43]. Because the monthly NDVI was around 0.5, the Chinese cork oak plantation had a relatively dense canopy in the canopy closure period, which was essential for the phenomenon where the LUE increased with a decreasing CI. When the ecosystem had a thin canopy, the $PAR_r$ could illuminate most leaves in the canopy, and the effects of the $PAR_f$ on the canopy's photosynthesis would consequently disappear [43].

The effects of biophysical factors on the LUE at the monthly time scale were different from those at the 30 min and daily time scales, as NDVI, an indicator of vegetation characteristics, was considered when analyzing the relationship between the biophysical factors and the LUE; similar to our study, vegetation growth had a positive effect on the LUE at the monthly time scale in various ecosystems [5,9,36]. The monthly PAR, $T_a$, VPD, P, and EF had positive effects on the monthly LUE in our study (Figure 9), which agrees well with the results in deciduous temperate forests and evergreen temperate forests [3,5,31]. Seasonal variations in the monthly PAR, $T_a$, VPD, P, and EF were similar to those in vegetation growth (Figure 8), which is the main reason for the relationship between the monthly PAR, $T_a$, VPD, P, and EF and LUE, because the vegetation growth determined the monthly GPP [12]. The partial correlation values between the monthly PAR, $T_a$, VPD, P, and EF and the LUE were negative or around zero (Figure 10b). In addition, the monthly CI was the sole factor influencing the LUE negatively (Figure 9), mainly because the CI influenced the quality of PAR received in the Chinese cork oak plantation, and the general trend in the monthly CI was opposite to that of the monthly GPP (Figure 9).

At the annual time scale, the LUE positively and negatively correlated with the GPP and PAR in our study, respectively (Figure 11a). Garbulsky et al. (2010) found that the annual LUE was positively linked with the annual GPP in global forest ecosystems [3], and Zhu et al. (2016) pointed out that the annual LUE had a negative effect on the annual LUE in the terrestrial ecosystems of China [26]. The annual LUE was negatively influenced by the annual CI ($p < 0.05$) in the Chinese cork oak plantation (Figure 11a), implying that the effect of the CI on the quality and quantity of the PAR still played a role in the photosynthetic efficiency at the annual time scale. Compared to other time scales, the relationships between the annual $T_a$, VPD, P, EF, and NDVI and the LUE were not remarkable at the annual time scale (Figure 11a), indicating that the same biophysical factors had different effects on the LUE at different time scales. According to the results of path analysis and partial correlation analysis, we found that that the quality and quantity of PAR and vegetation growth mainly

determined the LUE at different time scales in the Chinese cork oak plantation. In addition, according to the trends in climate change in our study area, the annual LUE of the Chinese cork oak plantation might decrease in the future. Improving management practices, such as fertilization [23] and irrigation [24], might slow or even reverse this trend in the annual LUE in the Chinese cork oak plantation.

## 5. Conclusions

In the examined Chinese cork oak plantation, diurnal LUE first increased sharply and reached a peak value at around 8:30, and thereafter decreased gradually until 12:00. It then increased gradually in the afternoon and reached its maximum value at sunset during the growing season. The general pattern in the daily LUE was similar to that of the monthly LUE, which first increased and then decreased within a year, and had a substantial drop around June. Great interannual variation was observed in the annual LUE, which ranged from 0.09 to 0.17 g C mol photon$^{-1}$, with a mean value of 0.12 g C mol photon$^{-1}$; while the multiyear mean LUE$_{max}$ was 0.30 g C mol photon$^{-1}$ during 2006–2019. Only GPP and CI had consistent effect on LUE at the 30 min, daily, monthly, and annual time scales, while the remaining biophysical factors had different effects on LUE at different time scales. The effects of the T$_a$, VPD, P, EF, and NDVI on the LUE were mainly indirect, via PAR and/or GPP. With an increasing CI, the PAR received by the canopy decreased, but the TD$_f$ increased, producing a more uniform irradiance in the canopy; this was the root cause for the LUE increasing with a decreasing CI. According to the climate change trends in the study area, the annual LUE might decrease in the future; improving management practices might slow or even reverse this trend in the annual LUE in this Chinese cork oak plantation.

**Author Contributions:** Conceptualization, X.G. and J.Z.; methodology, J.C. and P.M.; software, H.H.; validation, S.S.; writing—original draft preparation, X.G.; writing—review and editing, X.G. and J.Z.; visualization, X.G.; supervision, J.Z. All authors have read and agreed to the published version of the manuscript.

**Funding:** This research was funded by National Nonprofit Institute Research Grant of the Chinese Academy of Forestry (CAFYBB2022SY001) and National Natural Science Foundation of China (32301662).

**Data Availability Statement:** The data presented in this study are available on request from the corresponding author. The data are not publicly available due to the data needs to be used in many future works.

**Acknowledgments:** We appreciate editors and anonymous reviewers for their constructive suggestions and comments that improved the manuscript.

**Conflicts of Interest:** The authors declare no conflicts of interest.

## Appendix A

Interval: $0 \leq CI \leq 0.3$; Constraint: $S_f/S_e \leq CI$

$$S_f/S_e = CI[1.020 - 0.254CI + 0.0123\sin \beta] \tag{A1}$$

Interval: $0.3 < CI < 0.78$; Constraint: $0.1\,CI \leq S_f/S_e \leq 0.97\,CI$

$$S_f/S_e = CI[1.400 - 1.749CI + 0.177\sin \beta] \tag{A2}$$

Interval: $CI \geq 0.78$; Constraint: $S_f/S_e \geq 0.1\,CI$

$$S_f/S_e = CI[0.486CI + 0.182\sin \beta] \tag{A3}$$

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
