# Peer review of "Effects of Biophysical Factors on Light Use Efficiency at Multiple Time Scales in a Chinese Cork Oak Plantation Ecosystem"

_forests, doi:10.3390/f15091620_

Round 1
Reviewer 1 Report
Comments and Suggestions for Authors
This study analyzes the temporal variations in light use efficiency (LUE) over periods ranging from hours to years and examines how biophysical factors influence LUE at different time scales in a Chinese cork oak plantation using eddy covariance (EC) measurements. The article is structured according to the journal's requirements. However, the introduction must provide a comprehensive overview of the background on planted forests in this context. The research objectives are clearly articulated. The results are presented methodically, following the order outlined in the methodology section, and are clearly written and easy to understand. The discussion section appropriately references relevant literature and effectively elucidates the phenomena observed in the results. The methodology section contains all essential information and procedures.
Some minor revisions:
-It is necessary to describe what is known about LUE behavior in planted forests. Is it similar to native forests? What is the hypothesis about LUE behavior in planted versus natural forests? And specifically in cork oak?
-Path analysis is less common than correlation analysis. Please explain this statistical analysis in section 2.4.
-The caption of Figure 1 is incorrect. a) is Sr, b) Temp, and c) Prec. What do the numbers inside the figure represent? Is it the slope of the curve? Are the units in Sr correct?
-Line 138 – Does EddyPor calibrate the dataset?
-Line 143 – The soil heat flux and net radiation were not measured. How was the energy budget closure calculated?
-Line 172 – Is difficult to understand the equations. It´s necessary describes each step. Maybe is easier use a table with the equations or put tabs in the text. What is ‘q’? q=Sf/Si/CI = (Sf * CI) / Si?
-Line 198 – This behavior is observed not only in June but in all months except August, where there is no variation.
-Figure 2 – Are the LUE values at the beginning and end of the day consistent? Could this behavior be connected with the small PAR? Some papers start LUE analysis when PAR > 50 μmol/m²/s.
-Lines 231 to 233 – Rewrite this sentence: “Compared with daily PAR …” It is not clear, and the ideas are mixed.
-Figure 7 – The caption is confusing. I suggest: Relationship between CI and (a) GPP, (b)…
-“The partial correlation coefficient values between monthly Ta, VPD, P, EF, and NDVI and LUE were approximately zero at the significance level of p > 0.05, indicating that the effects of monthly Ta, VPD, P, EF, and NDVI on LUE were mainly via other biophysical factors (Fig. 10b).” What biophysical factors?
-How was the variance inflation factor calculated? It must be in the M&M section.
-Line 332 – The correct year is 1990. “As shown in Fig. 1, annual Sr increased by 9.96 MJ y⁻¹ (p = 0.064), annual mean Ta increased by 0.05 °C y⁻¹ (p < 0.001), and annual P increased by 4.28 mm y⁻¹ (p = 0.192) after 1990 in the study area.”
-The sentence: “Under the trends of climate change, the annual LUE of the Chinese cork oak plantation might decrease in the future, based on the results of correlation analysis among annual biophysical factors and LUE (Fig. 11a).” is a discussion, but it is not clear what variables are related.
-A reference is needed for this sentence: The main reason for this phenomenon is that vegetation uses relatively more chemical energy accumulated by earlier light reactions to absorb CO2, resulting in a slower decrease in GPP than PAR during the latter part of the day.
Reviewer 2 Report
Comments and Suggestions for Authors
The paper is focused on a rather popular problem in modern forest ecology, i.e. the study of the temporal variability of the Light Use Efficiency (LUE) of forest ecosystems. Chinese cork oak plantations were chosen as an example for the study.
LUE is widely used in remote sensing to determine the GPP of plant communities at regional and global scales. Eddy covariance CO2 flux measurements were used to calculate LUE, and this is the first point that requires additional explanation.
First, the methodology for determining GPP from measured flux data should be described in more detail. In fact, the authors used a commercial product in which they have full trust. However, it can sometimes be highly uncertain. The problem with modern algorithms, which are mainly based on the dependence of CO2 fluxes (RE, GPP) on temperature and solar radiation, is that they are rather approximate, with a large degree of error.
In addition, it is necessary to take into account the large number of missing data in the eddy covariance time series, which fall mainly during the night hours. By the way, the authors do not say anything about filling the gaps in the data. It is not clear whether the filled data were used in the determination of LUE or not? It can be expected that the coarseness of the algorithm used to determine the GPP is the reason for the LUE anomalies in the morning and evening hours. Another simple reason could be the error of the PAR sensor used to measure the incoming radiation at high solar zenith angles (measurements at low solar heights are a known shortcoming of this device).
Modern studies to fill gaps and estimate GPP and RE very often use machine learning techniques, which can also be applied in this study.
The description of the measurement design lacks a map showing the location of the study area and the footprint estimate. The description of the flux corrections used to determine fluxes by EC method should also be briefly presented in the methodology.
In addition, the reasons for using incoming PAR instead of absorbed PAR to determine LUE in the study are not entirely clear. Ultimately, the purpose of this LUE methodology is to verify remote sensing algorithms for determining GPP. And all of these remote sensing algorithms are based on absorbed PAR.
The next question relates to the result chapter with correlation analysis. When analyzing the data, the authors took the simplest way to determine the relationship between LUE and all possible parameters (influencing and non-influencing, dependent and independent). They did not even consider possible biological mechanisms affecting LUE. This is a formal approach that is not sufficiently scientific.
LUE defines the efficiency of vegetation to convert solar energy into biochemical energy through photosynthesis. It is not clear why the authors look for dependencies of LUE on GPP and PAR? For some reason they analyze and find relationships between obviously dependent parameters (e.g., PAR and CI). There's no logic to this.
Clearly, LUE depends on canopy stomatal conductance, which is controlled by biotic and abiotic factors such as incoming solar radiation, temperature, soil moisture, nutrient availability, etc. Precipitation affects soil moisture and can have short and long-term effects on plant functioning and stomatal opening. There is no direct effect of precipitation on LUE.
Therefore, I suggest a significant revision of the Results and Discussion chapters. With focus on biophysical control of LUE. And without strange relationships and regressions. Particularly the relationship between LUE and NDVI is strange. NDVI is a proxy for LAI, not LUE.
Of course, the analysis of daily and seasonal LUE dynamics is very interesting and needs to be considered in detail.
The manuscript requires additional English editing, preferably by a native speaker.
Specific comments.
L48
incoming PAR?
L54
plant density
L175-181
Eqs can be shifted in appendix.
L209
What do you mean under canopy stable period? I guess it is not correct term.
L212
What reason to look r between LUE and GPP?
L307
influencing ?
L412
“the canopy stable period” . What is it?
Comments on the Quality of English LanguageThe manuscript requires additional English editing, preferably by a native speaker.
Round 2
Reviewer 2 Report
Comments and Suggestions for Authors
The article has been revised to take into account the suggestions made, but it is still far from being accepted for publication.
Unfortunately, the main problem of the presented article is the lack of novelty and formally analyzed results.
The study uses quite interesting scientific data on the fluxes of sensible and latent heat and CO2 in a cork oak plantation ecosystem in China. New data on the temporal variability of light use efficiency are of great interest to readers. The data on the dependence of LUE on atmospheric and soil parameters are also very important. Unfortunately, the authors limited themselves to a general analysis of the temporal variability of LUE and the correlation coefficients. The main conclusion of the paper is the dependence of LUE on GPP and incoming PAR. This conclusion is not surprising, since this parameter (LUE) is defined as the ratio of these two parameters. It is difficult to expect anything else. Also the revealed dependence of LUE on the ratio of incoming solar and extraterrestrial irradiance at the upper boundary of the atmosphere is very obvious.
In my opinion, in addition to seasonal and interannual LUE dynamics, the main focus of the article can be placed on the assessment of LUE variability under the influence of solar radiation (without division into direct and diffuse), temperature, and soil moisture conditions. Special attention should be paid to the description of the biophysical mechanisms of LUE response to changes in external parameters. The response of LUE to extreme weather events may also be analyzed.
The article contains many stylistic and terminological errors. The article needs editing, preferably by a native English speaker.
Specific comments
L21-22 It is actually very obvious.
L57 I recommend avoiding the term carbon fluxes for the atmosphere. There are many constituents (gases, aerosols) in the atmosphere that consist of carbon molecules, including black carbon. In this case, please use CO2 and CH4.
L223 Influencing ?
L226 Graph shows diurnal variability of corresponding parameters...not daytime values.
L254 This graph illustrates only the obvious and well-known dependence of GPP on NDVI.
L296 The units for PAR are wrong.
It should be mol/m2day...not per second.
L297 In your study you have plotted the separate dependence of GPP on direct and diffuse PAR... Actually, such dividing makes absolutely no sense!
From my point of view, the dependence (figures) of daily GPP on PAR, maybe on mean daily temperature, soil moisture will be much more informative and useful.
L342 influencing?
L358 In fact, annual PAR can be expressed in MJ/m2 year.
L379 Not diurnal - hourly
L380 It is clear. There is no need to write about it in every sentence.
L393-394 Please provide an ecophysiological explanation.
L395 Please rephrase the sentence
L408 This is a phenomenon that needs to be studied in more detail in the study.
L434 In the summer months, when there is no defoliation?
L436 It is obvious, it is a very trivial conclusion.
L457 Again, this is very trivial ... as is the conclusion that the earth is round.
The amount of diffuse radiation is a function of atmospheric transparency and cloudiness.
L459 Please reconsider the sentence .... can take into account the difference in penetration of direct and diffuse radiation in the plant canopy.
L467 What about LAI?
L498 Absolutely trivial conclusion.
Comments on the Quality of English Language
Moderate editing of English language required.
